# Mathematical and computational modeling for organic and insect frass fertilizer production: A systematic review

Malontema Katchali[1,2], Edward Richard[1], Henri E. Z. Tonnang[2], Chrysantus M. Tanga[2], Dennis Beesigamukama[2], Kennedy Senagi[2]*

1 Institute for Basic Sciences, Technology and Innovation, Pan African University, Kenya, 2 Data Management, Modelling and Geo-Information Unit, International Centre of Insect Physiology and Ecology, Kenya

* ksenagi@icipe.org

**Data Availability Statement:** Data resources used to write this systematic review article are well referenced within the manuscript.

## Abstract

Organic fertilizers have been identified as a sustainable agricultural practice that can enhance productivity and reduce environmental impact. Recently, the European Union defined and accepted insect frass as an innovative and emerging organic fertilizer. In the wider domain of organic fertilizers, mathematical and computational models have been developed to optimize their production and application conditions. However, with the advancement in policies and regulations, modelling has shifted towards efficiencies in the deployment of these technologies. Therefore, this paper reviews and critically analyzes the recent developments in the mathematical and computation modeling that have promoted various organic fertilizer products including insect frass. We reviewed a total of 35 studies and discussed the methodologies, benefits, and challenges associated with the use of these models. The results show that mathematical and computational modeling can improve the efficiency and effectiveness of organic fertilizer production, leading to improved agricultural productivity and reduced environmental impact. Mathematical models such as simulation, regression, dynamics, and kinetics have been applied while computational data driven machine learning models such as random forest, support vector machines, gradient boosting, and artificial neural networks have also been applied as well. These models have been used in quantifying nutrients concentration/release, effects of nutrients in agro-production, and fertilizer treatment. This paper also discusses prospects for the use of these models, including the development of more comprehensive and accurate models and integration with emerging technologies such as Internet of Things.

## Introduction

Agriculture plays a critical role in global food production and the economy. However, it faces challenges such as population growth, climate change, and pandemics (e.g. COVID-19), which threaten food security [1]. Moreover, agriculture can have negative environmental impacts, including soil degradation, biodiversity loss, and greenhouse gas emissions, but sustainable

**Funding:** The authors gratefully acknowledge the financial support for this research by the following organizations and agencies: Australian Centre for International Agricultural Research (ACIAR) (ProteinAfrica –Grant No: LS/2020/154), Global Affairs Canada (BRAINS project: P011585), Novo Nordisk Foundation (ReflPro: NNF22SA0078466), the Rockefeller Foundation (WAVE-IN—Grant No: 2021 FOD 030); Bill and Melinda Gates Foundation (INV-032416); IKEA Foundation (G-2204-02144), European Commission (HORIZON 101060762 NESTLER and HORIZON 101136739 INNOECOFOOD), the Curt Bergfors Foundation Food Planet Prize Award; Norwegian Agency for Development Cooperation, the Section for Research, Innovation, and Higher Education grant number RAF–3058 KEN–18/0005 (CAP–Africa); the Swedish International Development Cooperation Agency (Sida); the Swiss Agency for Development and Cooperation (SDC); the Australian Centre for International Agricultural Research (ACIAR); the Norwegian Agency for Development Cooperation (Norad); the German Federal Ministry for Economic Cooperation and Development (BMZ); the Federal Democratic Republic of Ethiopia; and the Government of the Republic of Kenya. The funders had no role in study design, data collection and analysis, decision to publish, or preparation of the manuscript.

**Competing interests:** The authors have declared that no competing interests exist.

practices such as conservation agriculture and organic farming can mitigate these effects. The global agriculture market was valued at over 7 trillion U.S. dollars in 2019, providing employment, income, and food for approximately 9 billion people by 2050 [2]. Agriculture is the backbone of crop production and is critical for feeding the world's population and sustaining the global economy. According to the *Food and Agriculture Organization (FAO)* of the *United Nations*, agriculture employs over 1.3 billion people and contributes to around 10% of the world's gross domestic product (GDP) [1]. Poole et al. [3] noted that crop production is particularly important for providing food for humans.

Commercial fertilizers are made of synthetic or inorganic materials. Fertilizers may be customized to satisfy the precise nutritional needs of crops, but excessive usage might have a detrimental effect on the environment. A more environmentally friendly option is organic fertilizers, which are produced from natural materials like plant or animal manure. They enhance the health and fertility of the soil, diminish its negative effects on the environment, and increase the soil's ability to retain water. According to Zaccone et al. [4], micro-nutrients that may not be present in commercial fertilizers can also be provided by organic fertilizers. Organic fertilizers can be optimized during their process of production.

Mathematical modeling is a process of creating a mathematical representation of a system or process [5]. It is used to understand the underlying mechanisms and make predictions about the system's behavior under different conditions. Computational modelers computers to simulate the behavior of a system or process [6]. These techniques can enable farmers to determine the optimal amount, timing, and placement of fertilizers, improving crop yields while reducing environmental impact. In recent years, mathematical and computational tools such as simulation models, machine learning algorithms, and optimization techniques have been used to analyze the effectiveness of different organic fertilizers, evaluate their impact on soil fertility and crop yield, and assess their environmental sustainability. For instance, Xie et al. [7] used a dynamic modeling approach to optimize the use of organic fertilizers in tea plantation production. Their results showed that the optimized use of organic fertilizers led to significant improvements in soil fertility, tea yield, and economic benefits. Similarly, authors in [8] developed a comprehensive model to simulate the impact of organic fertilizers on wheat yield and nitrogen utilization efficiency. Moreover, machine learning algorithms have been used to predict the optimal amount of organic fertilizers required for crop production. For example in [9] a review of machine learning-based models to predict the optimum amount of cow manure required for agriculture production in Iran was done. Their model accurately predicted the optimal amount of cow manure, leading to increased crop yield and reduced environmental pollution. Ramezanpour and Farajpour [10] used artificial neural networks and genetic algorithms to forecast and improve banana fruit output in greenhouses while studying nitrogen, potassium, and magnesium as independent variables. Angouria-Tsorochidou and Thomsen [11] used a Monte Carlo [12] simulation type model to estimate the probability of exceeding the regulatory limits for heavy metal concentration in organic fertilizers. Crop yield and recommended fertilizer prediction using machine learning algorithms were implemented by Bondre and Mahagaonkar [13] to support farmers in their decisions making for appropriate fertilizer to use for crops. In order to identify the critical physical-mechanical parameters for the delivery of granular organic fertilizer, various research constructed numerical spreader models. For instance, in [14], a simulation model in the extended distinct element method (EDEM) program for the delivery of granular organic fertilizer with a centrifugal spreader was explored. It was determined that the model may be utilized for assessing the fertilizer's transverse homogeneity in the field (along the trajectory normal to the machine).

In 2021, the European Union (EU), Regulation 2021/1925 [15], defined frass as the insect excrement with amalgamation of parts of dead insects and feeding organic waste. This

regulation, adopts insect frass as an organic fertilizer in order to align its requirements for treatment and market as processed manure [15]. Insect-based organic fertilizers provide a sustainable and eco-friendly approach to enhancing soil fertility by converting organic waste into valuable nutrients, supporting plant growth, promoting a circular bioeconomy and sustainable agriculture [16, 17]. Among various methods, utilizing insects such as black soldier fly (BSF) has gained significant attention. Black soldier fly larvae (BSFL) are particularly efficient at transforming organic waste into high-quality fertilizers (within 5 weeks other than 24 weeks for conventional composting) making them a vital component in sustainable agriculture and animal feed [18, 19]. The integration of computational modeling has further enhanced the optimization of BSF-based frass fertilizer production, enabling precise predictions and improvements in nutrient conversion efficiency and overall impact on crop and the environment [18, 20–22]. However, research efforts are still at the infancy and require more attention.

Various computational methods have been developed to improve the quality and stability of organic fertilizers during production. However, to the best of our knowledge, no recent systematic literature review has been written and published to highlight applications of computational models in optimizing the production of organic fertilizers. Therefore, this article presents state-of-the-art literature on computational and mathematical methods for modeling the organic fertilizer production process.

In this article, the related works, methodology, and results are discussed in relation to the mathematical and computational modeling processes in organic fertilizer. Thereafter, the conclusion and future perspectives are presented.

## Literature review

### Previous works

The use of computational modeling in the context of organic fertilizer processes has advanced in a number of emerging domains, including the optimization of nutrients and organic fertilizer application. Ni et al. [23] established the key components of dissolved reactive phosphorus (DRP) losses from agricultural fields by applying multiple regression analysis to better understand the impacts of both site-specific conditions. They noted several context soil types and management practices e.g., agricultural conservation practices (ACP). It was identified that increased soil potential of hydrogen (pH), clay concentration, ACPs, winter cover crops, no-till and conservation tillage, fertilizer application, and precipitation all have an effect on DRP surface losses.

Lawrencia et al. [24] presented a literature review on the developments regarding controlled release fertilizers (CRFs). It was identified that the variables (temperature, pH, and ionic strength) significantly governed the nutrient release behavior. However, CRFs are continually developing and changing and there are other areas that are under investigation [25].

Sharma et al. [26] presented a systematic literature review on adaptive machine learning applications for sustainable agriculture supply chains (ASCs) performance. The study demonstrated how ML approaches may help ASCs and their sustainability. Terms that were discussed were crop yield, soil properties, and irrigation management. These constitute the pre-production and production phase. In the study, supervised, unsupervised, and reinforcement learning were all employed to create viable ASCs.

### Research gap and questions

In this study, a systematic review approach is carried out to examine how mathematical and computational modeling techniques have contributed to the optimal production and use of organic fertilizers in agriculture, their challenges/strengths, as well as their potential future

**Table 1. A list of research questions that guided this review.**

| Item | Research question (RQ) |
|------|------------------------|
| RQ1 | What are the assumptions and variables (independent and dependent) established by other researchers in the computational modeling of organic fertilizer to optimize agriculture production? |
| RQ2 | How are the computational models formulated, resolved, validated, and simulated? |
| RQ3 | What are the challenges or limitations of different computational models in the context of organic fertilizer production? |
| RQ4 | What recommendations can we provide for future research directions toward the production of organic fertilizer using computational models? |

trends. To achieve this, this article sought to address the following research questions as outlined in Table 1:

## Materials and methods

We adopted the Preferred Reporting Items for Systematic Reviews and Meta-Analyses (PRISMA) [27] methodology to derive and perform a meta-analysis of existing literature. The PRISMA methodology, shown in Fig 1, has four phrases, namely: identification, screening, eligibility, and inclusion as discussed below.

### Identification

This process focused on identifying relevant studies that discuss mathematical and computational models to optimize the production of organic fertilizers in agriculture, including insect-based frass fertilizer. A search, from January 2018 to March 2023, was conducted using the "advanced search function" of the literature's search database namely Google Scholar, Scopus, and Web of Science. Our search strategy included a combination of keywords listed in Table 2.

Given that the search databases were different from each other, a custom search equation was applied for each database as shown in Table 3. For Google Scholar, given its inclusion and the wide range of literature provided, and the challenges in automatic search filters, after applying this query, a manual identification was done to avoid multiple articles.

### Screening

In order to detect inconsistencies in the publications, the first screening process was done on the titles of the collected items in order to remove the reviews and articles whose titles did not meet our field of study. The second screening phase used the inclusion and exclusion criteria listed in Table 4. One of the procedures of screening involved reviewing the titles and abstracts, after which a comprehensive evaluation of the entire text of each sample was conducted, focusing on aspects like the clarity of the model and the kind of fertilizer on which the study was based. The results were stored in Mendeley software [28] in order to keep track of filtered articles and automatically extract metadata like author names, journal titles, and abstracts. These extracted pieces of information by Mendeley were stored in the JabRef software [29] to facilitate efficient organization, citation management, and seamless integration into the systematic review process.

### Eligibility

The bibliometric data were stored in a spreadsheet, which allowed for manual data entry, filtering, sorting, etc. Table 4 outlines the inclusion and exclusion criteria that determine the eligibility. Samples that met these criteria proceeded to the next stages of the study.

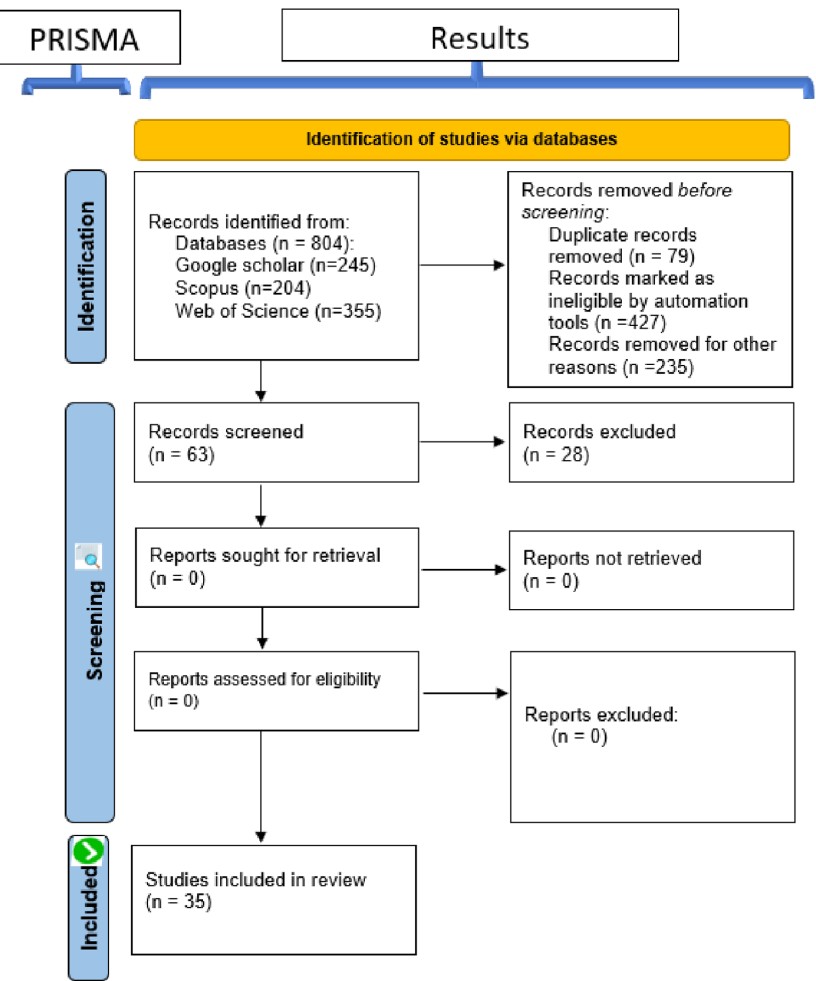

**Fig 1. A summary of the PRISMA steps to find the articles that were reviewed.**

**Table 2. Keywords for the study search.** The words marked with an asterisk (*) are the base/root words that can take various forms such as singular, plural, past, present, etc.

| Mathematical/Computational | Organic Fertilizer |
|---|---|
| Mathematical model* | Organic fertilizers |
| Computational model* | Manure* |
| Mathematical and computational model* | Compost* |
| Simulation model* | Biological fertilizer* |
| Dynamic model* | Biofertilizer* |
| Artificial intelligence | Green manure* |
| Machine learning | BSF |
| Statistical model* | Frass |
| Regression analysis | Insect frass |
| Multivariate analysis | |

**Table 3. Search queries.**

| Database | Search Query; from January 2018 to March 2023 |
|---|---|
| Scopus | (TITLE-ABS-KEY("Mathematical model*" OR "Computational model*" OR "Mathematical and computational model*" OR "Simulation model*" OR "Dynamic model*" OR "Artificial intelligence" OR "Machine learning" OR "Statistical model*" OR "Regression analysis" OR "Multivariate analysis") AND TITLE-ABS-KEY("Organic fertilizers" OR "Manure*" OR "Compost*" OR "Biological fertilizer*" OR "Biofertilizer*" OR "Green manure*" OR "Frass" OR "Insect Frass" OR "BSF") AND TITLE-ABS-KEY("Fertilizer application" OR "Agricultural production" OR "Sustainable agriculture" OR "Soil fertility" OR "Crop yield")) |
| Web of Science | (TS=(Mathematical model* OR Computational model* OR Simulation model* OR Artificial intelligence OR Machine learning OR Statistical model* OR Regression analysis OR Multivariate analysis) AND TS=(Organic fertilizers OR Manure* OR Compost* OR Biological fertilizer* OR Biofertilizer* OR Green manure* OR Frass OR Insect Frass OR BSF) AND TS=(Fertilizer application OR Agricultural production OR Sustainable agriculture OR Soil fertility OR Crop yield)) AND (OA = Yes OR OA = Open Access) |
| Google Scholar | "**With all of the words**" field: (title:("Mathematical model" OR "Computational model" OR "Mathematical and computational model" OR "Simulation model" OR "Dynamic model" OR "Artificial intelligence" OR "Machine learning" OR "Statistical model" OR "Regression analysis" OR "Multivariate analysis") OR abstract ("Mathematical model" OR "Computational model" OR "Mathematical and computational model" OR "Simulation model" OR "Dynamic model" OR "Artificial intelligence" OR "Machine learning" OR "Statistical model" OR "Regression analysis" OR "Multivariate analysis"))<br>"**With at least one of the words**" field: (title:("Organic fertilizers" OR "Manure" OR "Compost" OR "Biological fertilizer" OR "Biofertilizer" OR "Green manure") OR abstract: ("Organic fertilizers" OR "Manure" OR "Compost" OR "Biological fertilizer" OR "Biofertilizer" OR "Green manure" OR "Frass" OR "Insect Frass" OR "BSF")) |

**Inclusion.** Inclusion specifies relevant and included articles for further analysis. This ensures that the selected literature is relevant, of quality, and aligns with the research objectives of the review. After the preliminary screening, the qualified items were systematically analyzed. The analysis entailed examining the type of models used, assessing their assumptions, studying the key variables or parameters, reviewing challenges and limitations associated with them, and understanding the specific objectives tied to the modeling efforts within the broader research context.

**Method of assessing and evaluating risk of bias.** Assessing the risk of bias is crucial for evaluating the internal validity and reliability of the study results. To assess the risk of bias (ROB), a customized ROB framework (i.e., ROB-2) was used. ROB-2 is inline with the Cochrane Risk of Bias Tool [30–34] and is widely used for evaluating the risk of bias in

**Table 4. Inclusion and exclusion criteria.**

| Inclusion Criteria | Exclusion Criteria |
|---|---|
| Studies that focus on the use of mathematical and computational modeling techniques for organic fertilizer optimization in agricultural production | Studies that focus on non-organic fertilizer optimization in agricultural production |
| Studies published in peer-reviewed journals or conference proceedings | Studies that are not published in peer-reviewed journals or conference proceedings |
| Studies published in English language | Studies that are not published in English language |
| Studies published between 2018 and March 2023 | Studies published before 2018 |
| Presence of terms related to mathematical computational modeling | Studies that do not include mathematical or computational modeling techniques |
| Presence of terms related to agriculture: agronomy, agrarian, agriculture, and farm | Publications that do not address the role of mathematical andcomputational modeling in developing sustainable agricultural practices |
| Studies that present quantitative data and analysis | Studies that only present qualitative data and analysis |

randomized controlled trials (RCTs). In this article, the framework was used to assess bias across five dimensions, that is, assumptions, variables, formulation, resolution/validation, and simulation. These dimensions were chosen to address common biases in literature studies, focusing on aspects of computational and mathematical modeling approaches in organic fertilizer production. Two independent reviewers conducted the risk of bias assessments for each included study. The assessments were carried out independently to minimize bias and enhance the reliability of the results. figure.

### Risk of bias domains evaluated

With a focus on areas important to computational modeling in the context of the production of organic fertilizer, we evaluated the possibilities of bias in the papers that were ultimately chosen. Table 5 tabulates the bias assessment strategy for selecting articles to ensure their validity and dependability considering the research questions (set earlier in Table 1).

By evaluating the risk of bias across these domains specific to computational modeling, we ensured that our systematic review remains closely aligned with our research objectives and criteria. This approach allowed us to comprehensively assess the quality and reliability of the included studies and their relevance to the field of organic fertilizer production optimization. As shown in Table 5, an overall score of 8 or above indicates low risk of bias (well-selected) while a score below 6 suggests a higher risk of bias. This indicates the reliability of the selection process.

## Results

### PRISMA outputs

Fig 1 illustrates the PRISMA steps while Table 6 gives a detailed explanation. Generally, after conducting a comprehensive search of existing literature using the PRISMA methodology, the search gave 804 articles from Web of Science, Scopus, and Google Scholar search databases; in proportions illustrated by Fig 2. The first screening process yielded 63 articles and the second screening phase whittled those to 35. The remaining samples were deemed eligible and were included in the subsequent phases of the study.

### The major types of models

As shown in Fig 1, a total of 35 articles were collected using the PRISMA methodology. The articles that passed screening were then profiled and categorized into a) the mathematical and statistical models, and b) the data-driven models. Table 7 summarizes (11 articles) on data-driven models while Table 8 summarizes (24 articles) on mathematical and statistical models.

Statistical or mathematical-based modeling was used to study the production and nutrient dynamics of organic fertilizers [69–72]. It involved simulating the biological and chemical processes that occur during organic fertilizer production and predicting the changes in nutrient content and availability over time. Fig 3 illustrates the frequency of statistical and mathematical-based articles in the literature. Some of the models were RSM [35, 39], regression models [36], dynamic models [42], simulation models [14] and numerical models [40].

Computational modeling includes data-driven models such as machine learning and artificial intelligence methods. Data-driven modeling has been used in the context of organic fertilizer to forecast the efficacy of various fertilizer formulations and application rates depending on various environmental and soil variables [73]. These tools include Gaussian processes [57], non-linear support vector machine (SVM) and multi-layer perceptron (MLP) [64], extreme gradient boosting [56], particle swarm optimization (PSO) [60], artificial neural networks

**Table 5. Assessing risk of bias of the included papers.**

| Reference | Research Questions | | | | | Certainty Level/10 |
|---|---|---|---|---|---|---|
| | RQ1 | | RQ2 | | | |
| | D1 | D2 | D3 | D4 | D5 | |
| 1. Udume et al. [35] | + | + | X | + | X | 6 |
| 2. Attia et al. [36] | X | + | + | O | O | 6 |
| 3. Mucheru-Muna et al. [37] | X | + | + | O | O | 6 |
| 4. Krisnawati et al. [38] | X | + | + | + | + | 8 |
| 5. Asadu et al. [39] | X | + | + | + | O | 7 |
| 6. Bareha et al. [40] | X | + | + | + | O | 7 |
| 7. Zinkevičienė et al. [14] | X | + | + | + | + | 8 |
| 8. Xie et al. [41] | X | + | + | + | + | 8 |
| 9. Heinen et al. [42] | X | + | + | + | + | 8 |
| 10. Mebrate et al. [43] | X | + | + | + | O | 7 |
| 11. Schupp et al. [44] | X | + | + | O | + | 7 |
| 12. Singh et al. [45] | X | + | + | O | + | 7 |
| 13. Shang et al. [46] | X | + | + | + | X | 6 |
| 14. ZAILANI [47] | X | + | + | O | + | 7 |
| 15. Latif et al. [48] | X | + | + | + | + | 8 |
| 16. Walling and Vaneeckhaute [49] | + | + | + | + | O | 9 |
| 17. Tampio et al. [50] | X | + | + | O | + | 7 |
| 18. Zheng et al. [51] | X | + | + | + | + | 8 |
| 19. Matteau et al. [52] | X | + | + | O | O | 6 |
| 20. Jacob et al. [53] | + | + | + | O | + | 9 |
| 21. Li et al. [54] | X | + | + | + | X | 6 |
| 22. Dutta et al. [55] | X | + | + | + | X | 6 |
| 23. Towett et al. [56] | X | + | + | + | X | 6 |
| 24. Bui et al. [57] | X | + | + | + | X | 6 |
| 25. Genedy and Ogejo [58] | X | + | + | + | X | 6 |
| 26. Sharma et al. [59] | X | + | + | + | X | 6 |
| 27. Soto-Paz et al. [60] | X | + | + | + | X | 6 |
| 28. Spijker et al. [61] | X | + | + | + | X | 6 |
| 29. Kok et al. [62] | X | + | + | X | X | 4 |
| 30. Meng et al. [63] | X | + | + | + | X | 6 |
| 31. Pence et al. [64] | X | + | + | + | X | 6 |
| 32. Agyeman et al. [65] | X | + | + | + | X | 6 |
| 33. Guillaume et al. [66] | X | + | O | + | O | 6 |
| 34. Peng et al. [67] | + | + | + | + | O | 9 |
| 35. Wei et al. [68] | X | + | O | O | + | 6 |

| Key | | Judgement | | |
|---|---|---|---|---|
| D1: Assumptions | | Certainty | Certainty score | |
| D2: Variables | | + Inclusive | 2 | |
| D3: Formulation | | O Somehow | 1 | |
| D4: Resolution/Validation | | X Exclusive | 0 | |
| D5: Simulation | | | | |

**Table 6. Papers screening process.**

| Components | Google scholar | Scopus | Web of science |
|---|---|---|---|
| Search month | March, 2023 | March, 2023 | March, 2023 |
| Articles identified | n = 245 | n = 204 | n = 355 |
| Title screening and removing duplication | Manual screening (n = 245) | Using of search filters (n = 204) | Using search filters (n = 355) |
| Record excluded (n = 705) | Reviews/other languages/No specific type of fertilizer (n = 235) | No mathematical or computational terms/Not related to organic fertilizer (n = 174) | Reviews n = 36/ Unopened access/No mathematical or computational terms/Not related to organic fertilizer (n = 296) |
| Abstract screening and full text assessed for eligibility n = 63 | Article screened n = 10 | Article screened n = 30 | Article screened n = 23 |
| Record excluded after screening n = 27 | book(n = 1)/No clear modeling approach or model (n = 2) | Absence of clear model/Only experimental design/unopened access (n = 17) | Absence of clear model/Only experimental design/ unopened access (n = 8) |
| Record of articles that are eligible n = 35 | Meeting inclusion criteria: n = 7 | Meeting inclusion criteria: n = 13 | Meeting inclusion criteria: n = 15 |

(ANN), Landsat-8 image [62], etc. Fig 4 shows the frequency of articles that implemented data-driven models.

## Discussion

### Statistical and mathematical based modeling

Optimizing organic fertilizers has been realized in a variety of models. Different researchers have used mathematical and statistical modelling to comprehend different phenomena organic fertilizer production. Essentially, multiple factors must concurrently be taken into account when designing and simulating complex goods or processes [76]. This section is divided into two subsections: computational models in waste recycling and composting, and computational modeling for nutrient realization and soil fertility.

**Computational models in insect frass fertilizer.** Computational models have been applied in optimizing the efficacy and efficiency of insect-based fertilizers. Compared to other insects, BSFL have been widely used to generate high quality organic fertilizer. The studies [66–68] offer significant insights into several aspects of BSF-based fertilizer generation

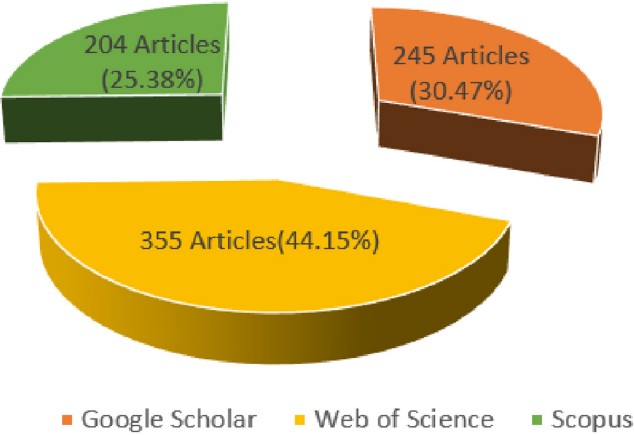

**Fig 2. Proportionality of number of papers found in each database search over the total papers identified.**

**Table 7. Summary of data-driven modeling for organic fertilizer.**

| Reference | Modeling | Model Purpose | Input Parameters | Target Variables | Data Source |
|---|---|---|---|---|---|
| Dutta et al. [55] | ANN (Artificial Neural Network), Random forest, Non-linear support vector machine | Forecast maize yield in smallholder agriculture system | pH, Nitrogen, Ec, Oc, Phosphorus, potassium, clay, sand, silt | Maize yield | Experimental data |
| Towett et al. [56] | Machine learning and non-destructive assays (Forest regression, Extreme gradient boosting) | Quantify the concentration of macro- and micronutrients | Compost, poultry manure, cattle and others | Nitrogen, Carbon concentration | Data from portable X-ray fluorescence (pXRF) and Diffuse Reflectance Fourier Transformed Mid-Infrared (DRIFT-MIR) spectroscopy |
| Bui et al. [57] | Data mining algorithm, Gaussian process | Predict nitrate and strontium concentration in groundwater | Temperature, pH, Ec, calcium, magnesium, and potassium ions | Nitrate and strontium concentration | Data acquired from [74, 75] |
| Genedy and Ogejo [58] | Gradient boosted trees, Random forest, Bagged tree, Neural network | Predict liquid dairy manure temperature | Time, manure depth, wind, solar radiation, humidity, rainfall, air temperature | Liquid dairy manure temperature | Data collected from on-farm manure storage |
| Sharma et al. [59] | ANN (Artificial Neural Network) | Predicting wheat grain yield | Nitrogen, potassium | Grain yield and productivity | Experimental data |
| Soto-Paz et al. [60] | Artificial neural network, particle swarm optimization | Optimization of biowaste composting | Mixing ratio, turning frequency, time | Compost quality | Experimental data |
| Spijker et al. [61] | Machine learning (Random forest) | Predict nitrate leaching from agricultural soils | Soil type, groundwater level, crop type | Nitrate concentration | Data from Dutch national LMM monitoring program |
| Kok et al. [62] | Landsat-8 images, and machine learning models | Classify macronutrients level in oil palm trees | Nitrogen, phosphorus, potassium, magnesium, and calcium from compost and inorganic fertilizer content | Macronutrients level | Data collected from five consecutive years of experiences |
| Meng et al. [63] | Random forest, adaptive boosting | Predict maize yield at the plot scale of different fertilizer system from 1994 to 2007 | Organic matter content, temperature, precipitation | Crop yield (maize yield) | Multi-source data (climate data, satellite data, fertilizer data, soil data) |
| Pence et al. [64] | Support vector machine, multi-layer perceptron, linear regression | Optimize energy and emissions from animal manure | Manure amount, number of animals, age | Biogas yield | Data from Turkstat |
| Agyeman et al. [65] | Hybridized empirical Bayesian kriging and support vector machine regression | Predict nickel concentration in peri-urban | Calcium, magnesium, potassium | Nickel concentration | Experimental data |

optimization. They employ a range of strategies to address issues such as compositional analysis, digestibility, processing methods, and sustainability. Guillaume et al. [66] implemented a logistic model for evaluating conversion efficiency yield. Comprehensive insights into digestibility and ideal larvae densities were revealed. Nevertheless, we note that validation of the model under diverse settings for generalization is needed. By improving mechanical parameters, Peng et al. [67] aimed to eliminate contaminants in BSF frass. Nevertheless, their methodology may not be applicable to other equipment and does not address the long-term operational feasibility or the economic implications of the optimized parameters. From these research, future studies can focus on adding more variables and automate the system. Though this approach may not cover all substrate varieties or real-world applications. Wei et al. [68] examined how different spent mushroom substrates and larvae affected frass quality. As a result, future research can broaden the range of substrates and incorporate complete models. Generally, the study point to the need for more comprehensive methods in order to improve sustainability and applicability, while also offering suggestions for enhancing the use of BSFL.

**Table 8. Summary of mathematical and statistical modeling for organic fertilizer.**

| Reference | Modeling | Model Purpose | Input Parameters | Target Variable (s) Related to Modeling Objects |
|---|---|---|---|---|
| Udume et al. [35] | Response surface method | Evaluation of the impact of composting factors on bio-degradation of lignin | Moisture content, inoculum size, turning frequency | Percentage lignin degradation |
| Attia et al. [36] | Switching regression model | Organic soil amendment and its effect on agriculture production | Osa adoption (binary), fertilizer type quantity | Wheat yield and net return |
| Mucheru-Muna et al. [37] | Multi-nominal regression model | Prediction of soil fertility for sustainable maize production | Age, yield, income, land | Integrated soil fertility quality |
| Krisnawati et al. [38] | Exponential polynomial model, logistic, linear regression | Identifying the response of spinach plants and exploring their growth | Rice husk bio-char, poultry manure content | Plants height, number of leaves, plants weight |
| Asadu et al. [39] | Response surface method | Optimization of process key parameters for bio-fertilizer synthesis in the soil | Composting time, dosage ratio, moisture content | NPK percentages |
| Bareha et al. [40] | Modified ADM1 model (Simulation model) | Predict the fate of Nitrogen during anaerobic digestion | Composite fraction $X_c$, polymer fraction $X_p$, particulate inert $X_1$, active biomass, substrate | C/N ratio, methane and ammonium productions |
| Zinkevičienė et al. [14] | Simulation model | Control granular fertilizer applications | Dry matter, particle density, granular diameter and length | Fertilizer concentration |
| Xie et al. [41] | Discrete element simulation model | Study the uniformity and stability of the fertilizer discharged from the feeder | Moisture content, bulk density, fertilizer volume, diameter | performance of organic fertilizer feeders |
| Heinen et al. [42] | Dynamic model | Prediction of the behavior synthesis fertilizer in the soil | Nitrogen and Phosphorus | Yield |
| Mebrate et al. [43] | Logistic regression model | Identification of determinants of soil fertility management practices | Enset plant, landholding, organic practices (yes or no) | soil fertility |
| Schupp et al. [44] | Long-term simulation model | Estimate the margin of exposure to Pb-induced toxic effects | fertilizer, ammunition deposition from air, Pb uptake, soil concentration, time, Pb blood level | Pb concentration in the soil, toxic effect, Pb uptake by plants |
| Singh et al. [45] | Crop dynamic model | Decision support system for N-fertilizer treatment | Nitrogen, biomass growth | Crop yield |
| Shang et al. [46] | Regression analysis | Yield trend and production evaluation | Nitrogen, phosphorus, potassium concentration | yield, rice production |
| ZAILANI [47] | Kinetic model | Determine the effect of moisture content on the properties of compost, predict the rate of composting | temperature, airflow, moisture, water content | compost, total organic carbon |
| Latif et al. [48] | Dynamic system model | Minimize waste and maximize resources | Organic waste, biomass | fertilizer production |
| Walling and Vaneeckhaute [49] | Kinetic model | Produce generalizable and simple composting system | temperature, moisture, oxygen content, time | compost, degradation rate |
| Tampio et al. [50] | Optimization model (Linear allocation model) | Obtain the most design for the digestate processing(Biogas plant investment decisions) | Consist of the characteristics of the case region in which the nutrient recycling is to be optimized (digested feed, nutrient efficiency, costs) | fertilizer production quality |
| Zheng et al. [51] | Integrated numerical and simulation model | Validation of organic soil amendment impact on farms performance | Farmyard manure contents | Wheat yield |
| Matteau et al. [52] | Coupled empirical nitrate production model | Prediction of nitrate leaching | Time, water content, temperature, $NH_4$, $NO_3$ | Fertilizer supply nitrogen concentration |
| Jacob et al. [53] | Spatial mathematical model (Finite element method) | Predict the variation in controlling parameters | Temperature, oxygen ratio, substrate, airflow | Compost quality |
| Li et al. [54] | Multi regression model | Organic amendment on dry land soil | Nitrogen, phosphorus, potassium, gram bacteria | Soil quality index |

(*Continued*)

**Table 8.** (Continued)

| Reference | Modeling | Model Purpose | Input Parameters | Target Variable (s) Related to Modeling Objects |
|---|---|---|---|---|
| Guillaume et al. [66] | Logistic model | Evaluate the Estimated digestibility (ED) of various predictive macronutrients in BSF and optimizing diet formulation | Larval density, type of feed, feeding time | Estimated digestibility (ED) of macronutrients, survival rate, feed conversion ratio (FCR), Larvae dry mass |
| Peng et al. [67] | Box–Behnken design, RSM | Optimize the operational parameters (trommel rotation speed, spike teeth rotation speed, and trommel inclination) to minimize impurity content in BSF frass mixtures | Rotational speed of trommel and spike teeth, Inclination angle of the trommel | Impurity content in BSF frass and rate of insect impurities |
| Wei et al. [68] | Regression analysis | Organic carbon dynamics analysis | Type of spent mushroom substrate (SMS), organic matter, brevitarsis larvae | Humic acid content, lignin utilization efficiency, phytotoxicity, fertilizer quality |

In summary, while dietary optimization and conversion efficiency are important, comparison analysis shows that practical processing methods and sustainability assessments offer complementary insights that improve our understanding of BSF systems as a whole. They bring to a wider understanding of BSFL as a practical and effective solution for waste management and fertilizer production by examining several facets of BSF systems.

**Computational models in organic waste recycling and composting.** Udume et al. [35] formulated the response surface method (RSM) model by considering three independent variables (moisture content, inoculum size, turning frequency), and the optimal conditions for maximum lignin degradation were determined. The statistical analysis was performed using the Design-Expert [77] software to resolve the model, and the model was simulated by performing experiments in triplicate under the optimal conditions, namely 65.7% moisture content, 7.5% inoculum concentration, and 5-day turning interval. Overall, the study demonstrated the potential of using RSM to optimize the bio-composting process for

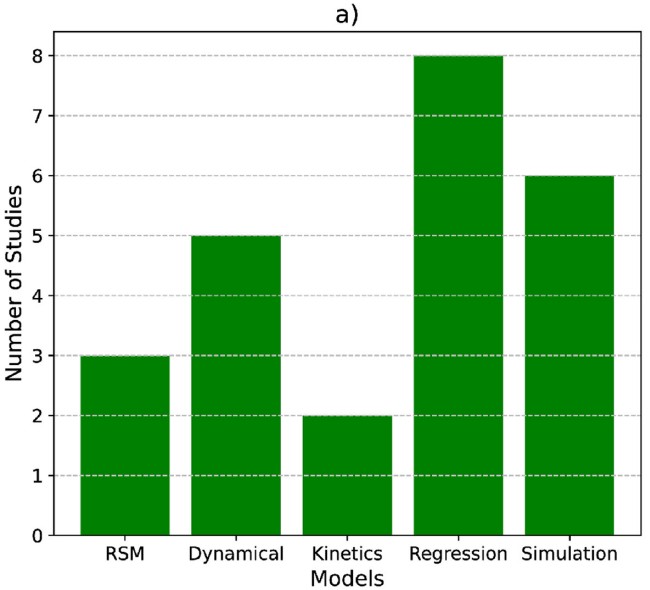
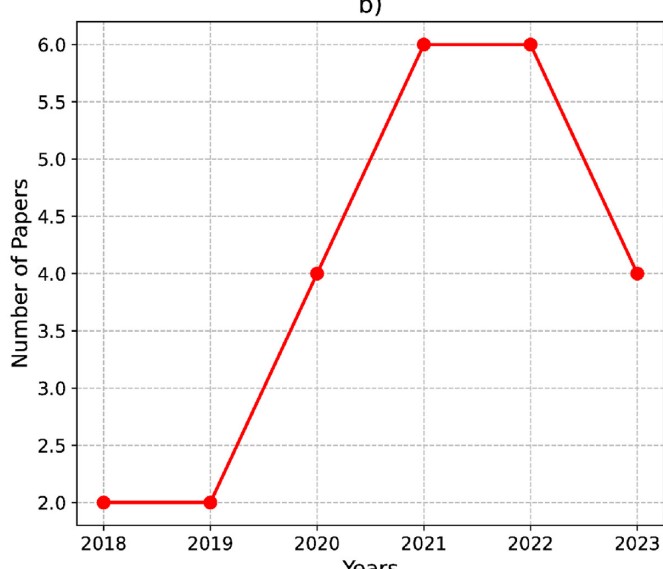

**Fig 3. Research models and publication trends using mathematical and statistical techniques.** a) Models used during studies, b) number of studies investigated per year using mathematical and statistical modeling approaches.

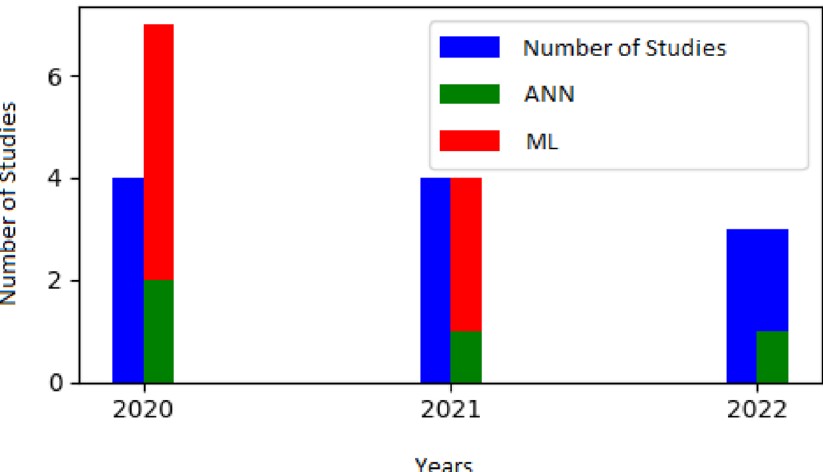

**Fig 4. This illustrates the number of studies that used Artificial Neural Networks (ANN) and Machine Learning (ML).** The blue bars represent the total number of studies eligible each year, while the green and red bars represent the number of studies specifically related to ANN and ML tools, respectively.

converting lignocellulosic biomass into a valuable bioproduct. The integrity and dependability of the model are shown by the $R^2$ score of 0.9733. As a consequence, the RSM method produced a fine-grained, dark brown Nutri-compost that was successful in increasing soil fertility. However, Asadu et al. [39] used the same approach (RSM) and validated their model by calculating the adjusted R-square ($R^2_{adj}$) defined in Eq (1). Regression coefficients of 98.60%, 99.79%, and 97.80% between anticipated and actual values for nitrogen (N), phosphorus (P), and potassium (K) show how the proposed regression models can interpret the experimental data. For both, no clear assumptions were stated. On the other hand, Heinen et al. [42] simulation of the fate of nitrogen and phosphorous was found by the convective dispersion-diffusion based on a non-linear partial differential equation that was resolved numerically. According to our examination, this was a predictive dynamical model. We note that [39] took into account the simultaneous behavior of the three nutrients (NPK) in their research, however, [55, 62] was data-driven. Nonetheless, Baligar and Bennett [78] noted that in a good fertilizer NPK content must be present in appropriate ratios.

$$R^2_{adj} = 1 - \frac{n-1}{n-q}(1 - R^2) \tag{1}$$

The optimization of organic fertilizer and the composting processes have been described using a variety of kinetic models (such as kinetics of biomass growth and substrate degradation models). Given that substrate degradation is typically the rate-limiting stage (the slowest step in a process that governs the overall rate of the entire process) [49, 79], the moisture content is frequently taken into account in the most fundamental composting models [47]. The Monod-type equations expressed in Eq (3), and first-order kinetic expression with correction functions (such as temperature correction, moisture corrections, air inflow corrections) to adjust for the impact of different process limiting variables defined by Eq (2) were the most prevalent expressions in a composting or substrate degradation modeling with $n$ being the number of experimental runs, and $q$, the quantity of indicators in the model. It was also assumed that the dependent variable $R_{degradation}$ which is the rate of degradation and $S_i$, the mass or concentration of biodegradable substrate, t, the time, $k_d$ the rate constant, and $k_T$, $k_{MC}$ and $k_{O_2}$ are non-

dimensional correction functions for temperature, moisture content and oxygen content respectively and expressed in Eq (2):

$$R_{degradation} = -\frac{dS_i}{dt} = k_d k_T k_{MC} k_{O_2} [S_i] \tag{2}$$

Nevertheless, in Eq (3), $\mu_i$ and $\mu_{max,i}$ are respectively specific and maximum growth rates of microbial species $B_i$, $\sigma_{M,i}$ is the Michaelis constant and $Y_{S_i}$ is the yield parameter of the quantity of biomass produced over the quantity of substrate consumed.

$$R_{degradation} = -\frac{dS_i}{dt} = \mu_i \frac{B_i}{Y_{S_i}} = \frac{\mu_{max,i}[S_i]}{\sigma_{M,i} + [S_i]} \frac{B_i}{Y_{S_i}} \tag{3}$$

Similarly, some considerations can be taken into account for the generalization of the model to the open and closed system of composting if some other factors (such as the input/output flow rate and the inlet substrate concentration) are considered in Eq (3). This adaptation involves incorporating these fluxes to account for the customary developmental process when modeling continuous art. It is analogous to an open system in which ongoing nutrient influx is maintained, and certain biological materials are concurrently removed from the arts. Martalò et al. [80] highlighted this weakness and yielded a system of equations including the modified Eq (3). For so doing with ($X$, $d$, and $K_s$) the biomass, death rate, and the substrate $S$ saturation coefficient respectively, with input substrate ($S_i$) and the biomass yield ($y_x$), the system (Eq (4)) showing the biomass growth in relation to its growing medium substrate.

$$\begin{cases} \dfrac{dX}{dt} = \left( \mu_{max} \dfrac{S}{K_s + S} - k_d \right) X \\[2mm] \dfrac{dS}{dt} = k_d \left( -\dfrac{X}{y_x} + S_i - S \right) - \dfrac{\frac{dX}{dt}}{y_x} \end{cases} \tag{4}$$

From this system, another critique point could be to include the anaerobic digestion process in the biodegradation of the organic matter induced by the action of biomass or bio-organism. Such a suggestion can include a new equation (Eq (5) with $\alpha$, the production coefficient) in the system 4 ($n = 2$ equations) describing the produced organic during the composting process over time.

$$\frac{dP}{dt} = -\alpha \frac{dS}{dt} \tag{5}$$

Approximately 15% of studies utilized one or more correction functions despite the extensive usage of correction functions [81]. This is due to the fact that mechanistic models make corrective functions less significant. Beyond this, a few studies have employed the spatial mathematical expression to account for the swarming of microorganisms in the substrate recycling in composting process [53]. In the latter case, the authors expressed a Monod type equation as the substrate correction function under the assumption that the substrate is homogeneous and homogeneously distributed, and the total volume and density of the compost, heat, and thermal are all constant. Standardized expression for substrate degradation is expressed in Eq (6) With $S_0$, the initial substrate quantity for composting.

$$h(S) = \frac{6}{5} \frac{S}{S + S_0/5}, h(S_0) = 0 \tag{6}$$

However, in this case, in addition to temperature correction, it has been involved that the model's oxygen ($k_{O_2}$, oxygen consumption rate) adjustment function (Eq (7)) is of the Monod type and relation between the half-saturation constant ($C_{o/2}$) for oxygen ($O_2$), the oxygen in the ambient air ($O_{2,amb}$) and the oxygen concentration ($xO_2$) in the pile can be presented as:

$$k_{O_2} = \frac{xO_2}{C_{o/2} + xO_2} \frac{C_{o/2} + O_{2,amb}}{O_{2,amb}} \qquad (7)$$

Substrate degradation results in temperature rise and oxygen consumption by microbes. The kinetics of substrate degradation (in Eqs (3) and (4)) offer an assessment of substrate deterioration with regard to time that is, in a sense, independent of earlier degradation, in contrast to approaches that use finite differences. The substrate dynamics model (time-varying model) can then be applied in integrated systems (anaerobic and aerobic processes) to observe the benefits of both approaches and can be applied for diversity in the substrate, opposite to the model proposed by Borisov et al. [82] which was for a single substrate.

In many processes and industries where exact management of substrate composting parameters (such as microbial communities, air inflow, C/N ratio) and the analysis of macro elements (nitrogen [N], phosphorus [P], and potassium [K]) [83] is necessary, correction functions for different environmental elements, such as moisture content, light, oxygen, humidity, and temperature, are vital. Correction functions for these environmental changes are essential in the context of composting in order to provide the perfect conditions for the efficient breakdown of organic materials [81, 84]. Arrhenius functions [85] formulated models of how temperature affects composting reaction rates. The challenge with organic degradation modeling, as stated in Eq (2), is that all of the corrective parameters change with time. For instance, the cardinal temperature equation (Eq (8)), the Haug equation for moisture (Eq (9)), and the Monod equation for oxygen (Eq (10)) were frequently utilized as the expressions for these corrections in the research of Walling and Vaneeckhaute [49].

$$f_T = \frac{(T - T_{\max})(T - T_{\min})}{(T_{opt} - T_{\min})[(T_{opt} - T_{\min})(T - T_{opt})}$$
$$-(T_{opt} - T_{\max})(T_{opt} + T_{\min} - 2T)]} \qquad (8)$$

$$f_{MC} = \frac{1}{\exp(-17.684MC + 7.0622) + 1} \qquad (9)$$

$$f_{O_2} = \frac{O_2}{O_2 + k_{O_2}} \qquad (10)$$

Composting models can be broadened by incorporating correction functions and kinetic data, such as degradation rates at reference temperatures, but this also increased the complexity of the models. The Arrhenius temperature function is used in composting to comprehend and forecast how temperature affects the breakdown process. One author tried to use the knowledge by expressing in two forms the temperature adjustment function. Since many microbial activities depend on temperature, we considered organic matter degradation. Therefore models have to account for temperature dependency. In that way, Heinen et al. [42] expressed temperature dependency using two approaches: *Van't Hoff* function (Eq (11)) and *Arrhenius* function (Eq (12)). Their performance with experiences showed that the two model approaches lead to a good prediction of nutrient leaches with a remark that this can be

quantitatively approximated, for example using Euler integration.

$$f_T = \frac{Q_{10}^{(T-T_{ref})}}{10} \tag{11}$$

$$f_T = A^{(T-T_{ref})} \tag{12}$$

The reliance of those correction functions is so crucial in such a way that most biological processes are environmentally dependent and so it is meaningful in the computational formulation of the scenarios. For example, Walling and Vaneeckhaute [49] brought forth a formulation of how substrate degradation by bi-organism is primarily related to the production unit temperature, oxygen diffusion, and substrate moisture. Nevertheless, Van't Hoff and Arrhenius functions in Heinen et al. [42] provided a good implementation of temperature effect on many bio-degradation and composting processes but when assuming that the oxygen content is supposed to be above 10%, and temperature above $25^o C$, this gives another facilitation to the actual model which is found to be incomplete or restricted.

**Computational models in nutrients released and soil fertility.** Nutrients play a vital role in soil fertility and when organic matter decomposes, it releases nutrients like nitrogen, phosphorus, and potassium into the soil, which can be absorbed by plants. However, soil fertility is affected by many factors other than just the presence of nutrients, including soil texture, pH levels, moisture, and temperature. Nonetheless, 40% of articles documented in Table 8, discuss nutrient content and soil fertility modeling. The RSM has been used to show how critical process parameters for converting agro wastes into bio-fertilizers may be optimized [39]. Despite the focus on bio-fertilizer modeling, this study did not come across literature that predict the precise end quality of organic fertilizer. Only four studies [39, 40, 42, 53] sought to use structural modeling to attempt to predict how organic fertilizer would develop qualitatively. In mechanistically derived models, Mucheru-Muna et al. [37] asked for more modeling objectives of variables to accurately show the content of NPK in the soil. Jacob et al. [53] analyzed compost maturity through spatial mathematical modeling, though this does not provide much information on the actual NPK quality of the compost in a manner similar to how nutrients accumulate when composted using a soil conditioner. Currently, the three most sophisticated conventional models (RSM, linear allocation model, coupled empirical nitrate production model) for nutrient transformation and loss are those formulated in [39, 50, 52]. In addition to the modified ADM1 (Anaerobic Digestion Model No. 1) simulation model for predicting nitrogen during anaerobic digestion [40], some physicochemical processes were also described. For example, Zheng et al. [51] incorporated organic soil amendment impact on farms performance by integrated numerical and simulation model as well Shang et al. [46] with regression analysis applied for yield trend and production evaluation. Nevertheless, it should be noted that estimating the parameters for commercial organic fertilizer requires careful consideration. Volume, mass, and moisture content are required to be the basic material characteristics of the fertilizer. Xie et al. [41] managed to study each of those parameters when investigating parameter calibration for the discrete element simulation model of commercial organic fertilizer. The Gaussian curve function specified in Eq (13) (with $y_0$, $w$, $A$ and $x_c$, constants) was used to do the analysis. Results indicated that the computer simulation model and the physical test had an $R^2$ of 0.988.

$$f(x) = y_0 + \left(\frac{A}{w\sqrt{\frac{\pi}{2}}}\right)\exp\left(-2\frac{(x-x_c)^2}{w^2}\right) \tag{13}$$

## Data-driven modeling

Generally, data-driven modeling involves using machine learning and artificial intelligence techniques to build models based on massive amounts of data sets [86]. In Table 7, ML tools were found to be the most widely used approaches for fertilizer modeling and optimization. Bui et al. [57] used temperature, the potential of hydrogen (pH), electrical conductivity (EC), calcium, magnesium, and potassium ions as input parameters to predict nitrate and strontium concentration in groundwater. Results showed that in addition to factors with high correlation coefficients, variables with low correlation should also be taken into account as inputs to produce realistic results. Still, a database of 246 groundwater samples was split into subsets for training and testing, using various inputs related to water quality. There are restrictions to take into account, even if the study offered insights into prediction accuracy. However, data quality and quantity issues were not thoroughly covered, and generalized to different areas and pollutants were not addressed. Aspects of the practical implementation and assessment measures were also absent. Enhancements in these areas would increase the study's performance and applicability. Analysis revealed that the Gaussian process (GP) algorithm outperforms well than random forest according to model validation. Dutta et al. [55] used pH, nitrogen, EC, organic carbon, phosphorus, potassium, clay, sand, and silt as input parameters to forecast maize yield to integrate socioeconomic and crop management factors. Three major modeling approaches were used namely ANN, random forest, and the non-linear support vector machine. ANN yielded the least (25%) declassification on validation samples. The target variables were also different with some studies focusing on predicting the concentration of specific nutrients [62] or compounds, while others aimed to predict crop yield [63] or bio-gas production [64].

In terms of the modeling techniques used, while some studies used relatively simple techniques such as linear regression [64], others used more sophisticated techniques such as gradient boosting or GPs [56–58]. Overall, a suitability comparison between the different techniques depends on the specific problem being addressed and the amount of data available. That being said, some studies did report superior performance of certain modeling techniques. For example, Dutta et al. [55] reported that the random forest algorithm outperformed both the artificial neural network and the support vector machine in forecasting maize yield. Similarly, Soto-Paz et al. [60] reported that the combination of artificial neural networks and particle swarm optimization yielded better results in optimizing biowaste composting compared to other techniques such as genetic algorithms or simulated annealing.

In summary, the choice of input parameters and target variables, as well as the selection of appropriate modeling techniques, are critical factors in developing accurate data-driven models for organic fertilizer. While there is no one-size-fits-all solution, the results of the different studies suggest that sophisticated techniques such as random forests and artificial neural networks may offer improved performance compared to simpler techniques, depending on the specific problem being addressed.

## Models validation

Model validation is done after model formulation and simulation steps. One of the most common methods of the regression model (such as those of organic fertilizer) validation is the MAPE (Mean Absolute Percentage Error) test. MAPE is a time series-related evaluation metric [87] defined in Eq (14), where $A_i$; $F_i$ and $n$ are designed to be real data, simulation outcome data, and the number of data points respectively. For example, Latif et al. [48] the MAPE validation approach revealed that, based on dynamics system simulation results and validation, the projection of organic fertilizer production in 2023 is 1256.79 tons while predicting that in

2027, it will be 1167.67 tons. Similarly, most of the studies used other validation or performance testing methods in regression models such as Root Mean Square Error (RMSE), Nash-Sutcliffe Efficiency (NSE), and Percent of Bias (PBIAS). According to Towett et al. [56], this method is accurate and gives a good performance test. A cross-validation experiment employing 33% of all organic amendment samples in that line, using RMSE, showed that the XGBoost model was the best. New things to maintain in this review are the fact that some studies used validation methods that capture the different varying various parameters in organic fertilizer management. Such methods include the relative importance factor (Eq (16)) and Mean absolute error (Eq (15)) that was applied on temperature [58] where $y_t$ was the measured manure temperature at time, $\hat{y}$ was the predicted manure temperature by the model, and $n$ is the number of data points in the dataset. The difference between the root mean square error (RMSE) of the model trained with all input variables and the model without the input variable $i$ is known as $\triangle RMSE_i$. We noticed that the performance of the model also depends on the numerical approach used to solve the problem or during the numerical simulations for most of the dynamical models. Rung-kutta (RK) method [88] together with optimal control [89] scenarios is also a better way to minimize errors between the model prediction and the real data. As an example comparing Petric and Mustafić [90], which has not used the optimal control in composting dynamism model, and Martalò et al. [80] that used optimal control coupled with gradient method [91] to minimize the deviation of the model prediction from the real data, it is seen that optimal control together with (RK) numerical method yielded a good performance and validation of the model. The main reason for using (RK) in this kind of problem is not only it allows for the numerical resolution of the dynamism based on ordinary differential equations, but also gives a better discretization (non-linear programming, quadratic programming) and appropriate choice of time steps. But, model accuracy depends also on the method used for parameter estimation, and in several problems of dynamism, it is advised to use the non-linear least squares method to handle non-linear relationships and provide flexibility in the models. Table 9 gives a summary of models and their corresponding validation methods.

$$MAPE = \frac{1}{n} \sum_{i=1}^{n} \mid \frac{A_i - F_i}{A_i} \mid \times 100\% \tag{14}$$

$$MAE = \frac{1}{n} \sum_{t=0}^{n} |y_t - \hat{y}| \tag{15}$$

$$RI(\%) = \frac{\triangle RMSE_i \times 100}{\sum \triangle RMSE_i} \tag{16}$$

**Table 9. Models validation method.**

| Models | Validation method |
|---|---|
| ANN/Kinetic model [49, 59] | Average NRMSE |
| Response surface model/Regression analysis model [39, 46] | $R^2/R^2_{adj}$ |
| Coupled empirical nitrate production model [52] | Nash-Sutcliffe Efficiency (NSE) |
| Modified ADM1 model [40] | $R^2_{Contois}/R^2_{1storder}$ |
| ML/SVM [53, 65] | RMSE/MAE |
| ANN/Forest regression/Extreme gradient Boosting/Data mining [9, 56] | MAPE/$R^2$/MAE/PBIAS |

## Challenges in organic fertilizer process modeling

Table 8 summarizes a variety of mathematical models, including regression models, dynamics models, kinetic models, and simulation models applied in optimizing the production of organic fertilizers. The commonly used model was the RSM [35, 39], which can be used to predict the response variable at different levels of the independent variables and to identify the optimal combination of independent variables that will produce the desired response. However, RSM also has some challenges in microbial culture [92]. For example, it assumes that the response variable is continuous and that the relationship between the independent variables and the response variable is linear or can be adequately represented by a second-order polynomial equation, this may be redoubtable to map with time in order to understand the evolution of nutrient content in a given organic fertilizer. While regression models have been widely used, as shown in Table 8, to optimize the use of organic fertilizers, they do have a number of drawbacks. The assumption that the relationship between the dependent variable and independent factors is linear is one of the complex issues. In practice, there may be more intricate patterns in the connection between organic fertilizer application rates and plant growth than in a straight line. Regression models also presuppose that the error factors are independent and regularly distributed as stated by Mucheru-Muna et al. [37]. These presumptions might be broken, resulting in biased parameter estimations and incorrect predictions. In order to describe more variables, more equations and parameters are required, leading to the complexity of models.

Kinetic models have been applied in several studies (such as [47, 49]). One of the challenges in Kinetic models in determining the rate constants (e.g. the rate constant governing the initial rapid release phase) that are used to describe the kinetics of nutrient release. This is due to the fact that the rate constants are influenced by a number of variables, including temperature, soil qualities, and moisture content, all of which can be challenging to monitor and change over time. The challenge of composting modeling is that, because these variables change throughout the process, all correction functions change as a function of time [49]. Optimizing the use of organic fertilizers presents a specific application of dynamics models [48], and it is not exempt from challenges. The complexity of the biological and chemical processes involved in nitrogen cycling presents a challenge in the development of a dynamic model for organic fertilizer optimization. The model must take into account variables including plant absorption rates, microbial activity, and the availability of soil nutrients. These interconnected processes can be challenging to adequately predict, particularly when several elements interact [93]. Large volumes of data on soil features, crop traits, and meteorological conditions are needed for organic fertilizer optimization. It can be difficult to validate a dynamic model since it is necessary to compare it with real crop yields, which might be affected by variables other than fertilizer application.

In developing a simulation model for organic fertilizer optimization, researchers have faced several challenges. One major challenge is the complexity of biological processes involved in composting, which made it difficult to accurately model the process and validating the model against real-world data can also be challenging due to variability in raw materials and nutrient content [94].

Data-driven modeling is an approach that has the potential to improve our understanding of organic fertilizer and its effects on soil health and crop yields. However, there are several challenges and limitations that can arise when using data-driven modeling in organic fertilizer. One of the main challenges is the limited availability of high-quality data on organic fertilizers, which can make it difficult to build accurate models [95]. The complexity and variability of organic fertilizers are also major limitations, as they can make it difficult to characterize and

model accurately [96]. In addition, the limited understanding of soil biology and the resource-intensive nature of data collection can also present challenges in the data-driven modeling of organic fertilizer. Another challenge in data-driven modeling for organic fertilizer is the variation in quality and properties of organic fertilizers derived from composting. The properties of the final product depend on several factors such as the type of feed used, the composting method employed, and the duration of the composting process. These factors may introduce variations in the nutrient content, pH, moisture, and microbial population of the final product, making it difficult to create generalized models [97]. Furthermore, the physical and chemical properties of compost can also affect its application rate and the potential environmental impact, thus highlighting the importance of accounting for these factors when developing models for organic fertilizers derived from composting. Moreover, as new technologies emerge, the integration of mathematical and computational models with these technologies can be a challenge. For example, incorporating data from sensors may require modifications to existing models.

## Computational modeling in other domains

Computational models, some of which are used to incorporate intra-extra cellular kinetics in biological processes have advanced in recent years. A case study of parameter estimation and learning from data for mathematical models of hepatitis C viral kinetics was built by Reinharz et al. [98]. Two categories of models were formulated: the standard ordinary differential equation (ODE) hepatitis C virus kinetic model (Biphasic Model) taking into account target cells $T$, infected cells $I$, and free virus $V$. Whereas a second model based on a fourth partial differential equation (PDE) was applied for intracellular viral (vRNA) kinetics (multiscale model). Analysis revealed that the ODE model is mathematically much simpler as compared to the PDE multiscale model despite the fact that the ODE model can be solved analytically when assuming that $T$ is a constant. Parameter sensitivity analysis was performed using Pearson's correlation [99] and revealed that the error in the multiscale model was higher than the one in the biphasic model. However, numerical methods can be applied to both models. For instance, the numerical method used to solve the ODEs and PDEs is not clearly shown in the methodology but based on parameter estimation and learning data, in the case of non-linear models, it's worth noting that the non-linear curve fitting method [100] could yield favorable results, especially when the accuracy of the initial guess is high. Nevertheless, it is advisable to use some robust techniques such as neural networks, and particle swarm optimization to estimate the parameters since these methods are capable of elucidating the interrelationships between variables, thereby contributing to a more comprehensive understanding of the underlying system. That being said, Kalmar filter [101] is also among the best methods to accurately operates in parameters estimation compared to least square methods based on a mathematical model that represents the dynamic behavior of the system and the measurements obtained from the system by minimizing the discrepancy between the model predictions and the observed measurements. For example, Godinez and Rougier [102] presented dynamic combined finite discrete element methods using the Kalman Filter ensemble to simulate the evolution of fractures and cracks in different geo-materials. The findings demonstrate a continuous convergence of the assimilated parameter values towards the time/stress curves found from the observed data.

In the context of modeling techniques, it's essential to consider the methodology to be applied to a specific problem. For example, Chen et al. [103] investigated a new mathematical modeling approach for optimizing the dynamic response of the four-stage helicopter main gearbox. In that paper despite the wide use of some numerical methods such as the Euler method, the authors made use of the Fourier series method to solve the differential equations

of the system. After simulations and calculation of the dynamic response of each meshing element, results showed that the sensitivity analysis method may also be utilized to effectively select the optimal shaft node and pick the right parameters to lower the system response. However, the method (Fourier series method) of solving the ODEs was not really detailed in the process.

Some numerical approaches can be applied directly in comparison to the actual mathematical expression of the system being modeled. And so when applying methods such as Fourier and Laplace transforms [104, 105], this must be clearly shown the relationship to the problem being investigated. This refers us to the study of Farah et al. [106] with the fractional Fourier transform (FRFT) which is a time–frequency distribution and an extension of classical Fourier transform application in cryptanalysis and Laplace transform in cryptology [107]. In that area of study, the work of Sambas et al. [108] used the dynamical modeling approaches in mathematical modeling and field-programmable gate array (FPGA) realization of a multi-stable chaotic dynamical system with a closed butterfly-like curve of equilibrium points. Their work gave all components of dynamics such as stability tests by the use of the Lyapunov method [109]. The study gives an idea about the bifurcation, but it has not given a precise type such as pitchfork, trans-critical, or saddle-node as proposed in [110].

## Future research directions in computational modeling for organic fertilizer

### Emerging areas and areas requiring progress

Organic farming and the use of organic fertilizers have gained popularity in recent years due to growing concerns about the environmental impacts of conventional agriculture practices. Organic fertilizers are derived from natural sources and offer several benefits, including improving soil health and promoting sustainable agriculture. However, there are still emerging areas and areas requiring progress in the case of organic fertilizers, such as the standardization of production methods and improving nutrient availability. By addressing these issues, the organic fertilizer industry can continue to evolve and contribute to more sustainable agriculture practices.

There is a lack of standardization in organic fertilizer production, which makes it challenging for farmers to choose the right type of organic fertilizer for their crops. The development of standardized production methods and labeling guidelines can help ensure the quality and safety of organic fertilizers. Organic fertilizers often contain lower levels of nutrients than synthetic fertilizers, and the nutrients may not be immediately available to plants. Developing techniques to improve the availability of nutrients in organic fertilizers can help maximize their effectiveness. Organic fertilizers can also have environmental impacts, such as nutrient runoff and greenhouse gas emissions. More research is needed to understand the environmental impacts of organic fertilizers and to develop methods to mitigate them. For example, algae are a rich source of nutrients and can be used to produce organic fertilizers. Algae-based fertilizers have the potential to provide a sustainable source of nutrients for crops while reducing dependence on synthetic fertilizers [111]. Overall, these emerging areas offer opportunities for the use of mathematical and computational modeling in organic fertilizer production and application, leading to more sustainable agricultural practices and improved crop productivity. Given that organic fertilizer production require a lot of control due to the variability of the state vector that is entering the process, it is also advisable to use the controller's models such as proportional-integral-derivative (PID) controller [112], which is generally accepted, easy to use, and computationally effective. It frequently functions in a variety of applications, such as temperature regulation. Similar to PID, adaptive model predictive control (AMPC) [113] is an

integrating system model online adaption. It allows the controller to adjust to changes in the system dynamics or uncertainties by continually updating the model parameters based on in-the-moment data. This adaptive nature makes it possible to regulate dynamic systems more effectively and robustly. Note that these techniques have been used in chemical and electrical manufacturing, however to the best of our knowledge, none of the studies have employed these techniques to control inputs and state variables in the domain of organic fertilizer production which most of the time is produced in greenhouses. Unlike simple prediction models, these models allow to handle constraints such as boundary conditions and horizon time prediction making it suitable for a wide range of applications in various sectors.

The incorporation of numerous data sources, variability accounting, soil biology, the development of prediction models, and uncertainty incorporation are future goals for organic fertilizer. These characteristics can aid in the development of more precise and reliable models for forecasting the impact of various organic fertilizer types on soil health and crop yields. We may better comprehend the intricate relationships between many elements that affect soil health and crop yields by using cutting-edge methods like probabilistic modeling and machine learning. On the other hand, there are several reasons why mathematical and computational modeling in organic fertilizer optimization requires progress. One of the main challenges is the availability of high-quality data to develop and validate accurate models. Models can also be complex and require significant computational resources, making them less accessible to small-scale farmers and researchers. Furthermore, models must be validated against real-world data to ensure their accuracy and reliability. Additionally, models developed for specific organic fertilizer production and application systems may not be transferable to other systems. To ensure their relevance and applicability in real-world contexts, there is a need to incorporate socio-economic factors that influence organic fertilizer production and application into the models. The use of remote sensing and IoT technologies, cloud computing, and integration with other models such as climate and soil models can also improve the accuracy and applicability of these models. Training and capacity-building programs are needed to equip farmers and researchers with the necessary skills to develop and use these models, and government policies can play a significant role in promoting their development and use.

## Perspectives of advances in modelling insect frass fertilizer

Another knowledge gap we would like to highlight is the prediction of the biodegration of organic waste by insects to generate organic fertilizer. Recycling organic waste to produce organic fertilizer is becoming increasingly popular due to its environmental and economic benefits [114]. To develop models for organic fertilizer derived from recycled organic waste, it may be necessary to account for the type and quality of the organic waste used as feedstock, the processing method employed, and the duration of the composting process. Additionally, it may be useful to incorporate information on the initial nutrient content and quality of the organic waste. Many composting models [60] are only relevant to the waste sources they were designed and calibrated for due to the inability to generalize the process kinetics. This important shortcoming, which currently afflicts the field, must be addressed. Machine learning techniques can be used to optimize the composting process and improve the quality and consistency of the final product.

Recycling organic waste using insects such as BSF is an innovative and sustainable approach that has gained increasing attention in recent years [16, 66]. To develop models for organic fertilizer derived from insect-based systems, it may be necessary to account for factors such as the species of insects used, the environmental conditions in which they are raised, and the feedstock provided. Additionally, it may be useful to incorporate information on the nutritional

content and properties of the insect-derived organic fertilizer. Mathematical, statistical, and machine learning techniques can be used to optimize the insect-rearing process and improve the quality and consistency of the final product. Finally, it can be possible to develop predictive models that can help to optimize the use of insect-derived organic fertilizer in agriculture. In that way, Anyega et al. [115] looked to investigate the comparative performance of composted BSFL frass [67] fertilizer to other types, but the deficient of the study is that there is no consideration of the dynamism behind BSF, thereby to consider or optimize their capability to recycle waste and furthermore suggest a dynamical model that incorporates their lifestyle and waste recycling. Additionally, future research should focus on incorporating socio-economic factors and other relevant environmental factors into the models, as well as exploring the potential of emerging technologies such as remote sensing and IoTs in data collection [19]. Ultimately, the choice of modeling method should be validated against real-world data to ensure their accuracy and reliability. In addition, it is relevant to account for nonlinear model predictive control (NMPC) [116] which generates control actions that optimize a given objective function over a potential time horizon using nonlinear system models and optimization techniques. Nonlinear dynamics may be controlled using NMPC, which also handles state restrictions and control inputs including temperature, moisture content, and gas behavior. For example, the study of Padmanabha et al. [117] is a concrete utilization of model predictive control (MPC) in the biology sector.

## Limitations of the review processes

In the course of conducting this systematic review of mathematical and computational modeling in organic fertilizer production, we encountered several inherent limitations in our review processes.

1. Publication bias: Despite our best efforts to conduct a comprehensive search across multiple databases, perhaps some relevant studies were inadvertently omitted.

2. Language limitation: Our review was constrained to studies published in the English language. This language restriction may have led to the exclusion of valuable research published in other languages, introducing a potential bias by overlooking contributions from non-English-speaking researchers.

3. Database selection bias: We made use of specific databases (Scopus, Web of Science, and Google Scholar) to search for relevant articles for this study. However, our choice of the specific databases for the literature search might have unintentionally excluded other articles.

4. Search strategy limitations: Despite our rigorous search criteria, the inherent variability in terminology and indexing across different databases may have led to the inadvertent omission of relevant studies. Variations in keyword usage and indexing practices could have impacted the comprehensiveness of our search.

5. Time-frame limitation: Our review is confined to studies available up to a specific cutoff date. It is possible that newer research developments have emerged after this date, potentially impacting the relevance and completeness of our review.

6. Full text accessibility: Some studies may not have been included in our review due to restricted access to full texts. This limitation could affect the comprehensiveness of our review, as relevant information might have been missed.

In summary, despite these acknowledged limitations, our systematic review represents a comprehensive effort to synthesize the available literature on mathematical and computational modeling in organic fertilizer production. Recognizing these limitations is essential for a comprehensive interpretation of the findings, and we recommend that future research endeavors in this area aim to address these challenges to further enhance the understanding of this vital field.

## Conclusion

This review provides a consolidated view of the state of optimization and modeling methods or approaches for the improvement of organic and insect frass fertilizer production and applications. It was found that kinetics models are mostly used to explain the chemical and/or environmental variation in composting dynamism and the application of adjustment functions such as temperature, and moisture content to modify the kinetics. The representation of degradation kinetics, which has involved either first-order or Mond-type kinetics, has generated more debate than the widely accepted mass and heat balances. Although Monod-type kinetics should provide the most realistic results and there is some evidence to support this, both techniques have had good success. Almost half of the evaluated literature included correction functions in some capacity, whereas the other half completely disregarded their application because of the rise of mechanistic models. However, on the other hand, sophisticated techniques such as random forests and artificial neural networks may offer improved performance compared to simpler techniques. Although challenges such as the need for accurate and comprehensive data and model validation exist, the benefits of using these models in improving agricultural productivity and reducing environmental impact are significant. This systematic review emphasizes how advances in mathematical and computational models have led to improvements in crop yields and waste recycling efficiency by simulating biological processes and optimizing nutrient content in the creation of organic and insect frass fertilizer. Moreover, it emphasizes the necessity of multidisciplinary (e.g., mathematicians, computer scientists, and agronomists) cooperation for the advancement of this sector. Future prospects for the use of mathematical and computational models in organic and insect frass fertilizer production and application include the development of more comprehensive and accurate models and integration with emerging technologies. Overall, this review underscores the importance and the role of mathematical and computational modeling and suggests robust models such as nonlinear model predictive control, proportional-integral-derivative, and model predictive control to implement for effective control.

## Supporting information

**S1 File. Documentation of the results of the literature search, that was used to write this systematic review, can be accessed here—https://github.com/icipe-official/Systematic-Review-Modeling-for-Organic-and-Insect-Frass-Fertilize.**
(XLSX)

## Author Contributions

**Conceptualization:** Malontema Katchali, Chrysantus M. Tanga, Kennedy Senagi.

**Data curation:** Malontema Katchali, Edward Richard, Chrysantus M. Tanga, Kennedy Senagi.

**Formal analysis:** Malontema Katchali, Edward Richard, Chrysantus M. Tanga, Kennedy Senagi.

**Funding acquisition:** Chrysantus M. Tanga.

**Investigation:** Malontema Katchali, Edward Richard, Henri E. Z. Tonnang, Chrysantus M. Tanga, Kennedy Senagi.

**Methodology:** Malontema Katchali, Chrysantus M. Tanga, Dennis Beesigamukama, Kennedy Senagi.

**Project administration:** Chrysantus M. Tanga, Kennedy Senagi.

**Resources:** Malontema Katchali, Chrysantus M. Tanga, Kennedy Senagi.

**Software:** Malontema Katchali, Kennedy Senagi.

**Supervision:** Edward Richard, Chrysantus M. Tanga, Kennedy Senagi.

**Validation:** Malontema Katchali, Edward Richard, Henri E. Z. Tonnang, Chrysantus M. Tanga, Kennedy Senagi.

**Visualization:** Malontema Katchali, Kennedy Senagi.

**Writing – original draft:** Malontema Katchali, Kennedy Senagi.

**Writing – review & editing:** Malontema Katchali, Edward Richard, Henri E. Z. Tonnang, Chrysantus M. Tanga, Dennis Beesigamukama, Kennedy Senagi.

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
