## [Decision Letter · Decision Letter 0]

19 Jul 2024

PONE-D-23-28302Mathematical and Computational Modeling in Organic Fertilizer Production: A Systematic ReviewPLOS ONE

Dear Dr. Senagi,

Thank you for submitting your manuscript to PLOS ONE. After careful consideration, we feel that it has merit but does not fully meet PLOS ONE’s publication criteria as it currently stands. Therefore, we invite you to submit a revised version of the manuscript that addresses the points raised during the review process.

Please see the comments from three reviewers below.

We look forward to receiving your revised manuscript.

Kind regards,

Hanna Landenmark

Staff Editor

PLOS ONE

“The authors gratefully acknowledge the financial support for this research by the

following organizations and agencies: the Swedish International Development

Cooperation Agency (Sida); the Swiss Agency for Development and Cooperation (SDC);

the Australian Centre for International Agricultural Research (ACIAR); the Federal

Democratic Republic of Ethiopia; and the Government of the Republic of Kenya. The

views expressed herein do not necessarily reflect the official opinion of the donors.”

“The authors gratefully acknowledge the financial support for this research by the following organizations and agencies: the specific restricted project donor (written out in full) and grant number; the Swedish International Development Cooperation Agency (Sida); the Swiss Agency for Development and Cooperation (SDC); the Australian Centre for International Agricultural Research (ACIAR); the Norwegian Agency for Development Cooperation (Norad); the Federal Democratic Republic of Ethiopia; and the Government of the Republic of Kenya. The views expressed herein do not necessarily reflect the official opinion of the donors.”

4. We note that your Data Availability Statement is currently as follows: [All relevant data are within the manuscript and its Supporting information files]

Reviewers' comments:

Reviewer's Responses to Questions

**Comments to the Author**

1. Is the manuscript technically sound, and do the data support the conclusions?

Reviewer #1: Yes

Reviewer #2: Yes

Reviewer #3: Yes

2. Has the statistical analysis been performed appropriately and rigorously? 

Reviewer #1: Yes

Reviewer #2: N/A

Reviewer #3: Yes

3. Have the authors made all data underlying the findings in their manuscript fully available?

Reviewer #1: Yes

Reviewer #2: Yes

Reviewer #3: Yes

4. Is the manuscript presented in an intelligible fashion and written in standard English?

Reviewer #1: Yes

Reviewer #2: Yes

Reviewer #3: Yes

5. Review Comments to the Author

Reviewer #1: Dear authors,

I would like to extend my sincere congratulations to the authors for their exceptional work on this systematic review. The article well-written, and presenting the valuable from previous studies. The efforts in summarizing and critically evaluating the included studies have provided readers with valuable insights into the current state of knowledge in this field.

I have a few comment to be amended from your manuscript:

1) Page: 4, Table 2:

What is the meaning of symbol “*”?

Maybe the author can mention as foot note below the table regarding the meaning of “*”.

2)The references format inconsistent. Kindly check the format of manuscript provided by PLOS ONE.

I hope that your findings will guide future research efforts. Once again, congratulations on your outstanding systematic review.

Reviewer #2: Thank you so much for doing this attractive systematic review.

It has been written according to the PRISMA writing standard. However, once again, please check the manuscript in terms of heading and subheading with the PRISMA checklist.

In the discussion section, I am not sure whether the key findings is necessary or not. I think it is better to adhere to the journal`s instruction for authors and omit it if possible. Conclusion is enough.

Reviewer #3: As the statistical reviewer I will focus on methods and reporting. Overall, this is a well conducted and reported study and I only have a few points to raise.

Major

1) it was not clear if the authors searched thereference lists of the identified papers, for any relevant papers that were not returned by their searches. it is customary to do that and also report on the numbers and percentages missed. A low percentage is an indication of a good / working well search strategy.

2) I have concerns about the risk of bias tool used, it is not clear which one it is. is it ROB-2? this tools is specifically for RCTs and it is not clear how the domains map and are relevant to the studies included in this review. I'm not even sure an appropriate risk of bias tool exists, since there is no comparison really. Anyway, the authors need to reflect and explain in the next iteration of the paper.

Minor

1) meta analysis is mentioned a couple of times in the paper, I suggest not mentioning it since it is irrelevant (nothing to pool).

6. PLOS authors have the option to publish the peer review history of their article (what does this mean?). If published, this will include your full peer review and any attached files.

Reviewer #1: No

Reviewer #2: **Yes: **Mahin Nomali

Reviewer #3: No

---

## [Decision Letter · Decision Letter 1]

27 Sep 2024

Mathematical and Computational Modeling for Organic and Insect Frass Fertilizer Production: A Systematic Review

PONE-D-23-28302R1

Dear Dr. Kennedy Senagi

We’re pleased to inform you that your manuscript has been judged scientifically suitable for publication and will be formally accepted for publication once it meets all outstanding technical requirements.

Kind regards,

Noé Aguilar-Rivera

Academic Editor

PLOS ONE

---

## [Editor Report · Acceptance letter]

31 Dec 2024

PONE-D-23-28302R1 

PLOS ONE

Dear Dr. Senagi, 

I'm pleased to inform you that your manuscript has been deemed suitable for publication in PLOS ONE. Congratulations! Your manuscript is now being handed over to our production team.

Kind regards, 

on behalf of

Dr. Noé Aguilar-Rivera 

Academic Editor

PLOS ONE